# Targeting WEE1 Kinase for Breast Cancer Therapeutics: An Update

**DOI:** 10.3390/ijms26125701

**Published:** 2025-06-13

**Authors:** Zhao Zhang, Ritika Harish, Naveed Elahi, Sawanjit Saini, Aamir Telia, Manjit Kundlas, Allexes Koroleva, Israel N. Umoh, Manpreet Lota, Meha Bilkhu, Aladdin Kawaiah, Manogna R. Allala, Armelle Leukeu, Emmanuel Nebuwa, Nadiya Sharifi, Anthony W. Ashton, Xuanmao Jiao, Richard G. Pestell

**Affiliations:** 1Pennsylvania Cancer and Regenerative Medicine Research Center, Baruch S. Blumberg Institute, 100 East Lancaster Avenue, LIMR R234, Wynnewood, PA 19096, USA; 2Xavier University School of Medicine at Aruba, Woodbury, NY 11797, USA; 3Lankenau Institute for Medical Research, Wynnewood, PA 19096, USA; 4HUN-REN Office for Supported Research Groups, Cell Cycle Laboratory, National Institute of Oncology, 1122 Budapest, Hungary; 5Chemistry Coordinating Institute, University of Debrecen, 4012 Debrecen, Hungary; 6Semmelweis University, 1117 Budapest, Hungary; 7The Wistar Institute, Philadelphia, PA 19107, USA

**Keywords:** WEE1 kinase, WEE1 inhibitors, breast cancer, cyclin-dependent kinases, cell cycle regulation, G2/M checkpoint, mitotic entry, DNA damage/repair, chemotherapy resistance, clinical trial

## Abstract

WEE1 kinase is a crucial cell cycle regulatory protein that controls the timing of mitotic entry. WEE1, via inhibition of Cyclin-dependent Kinase 1 (CDK1) and Cyclin-dependent Kinase 2 (CDK2), governs the G2-M checkpoint by inhibiting entry into mitosis. The state of balance between WEE family kinases and CDC25C phosphatases restricts CDK1/CycB activity. The WEE kinase family consists of WEE1, PKMYT1, and WEE2 (WEE1B). WEE1 and PKMYT1 regulate entry into mitosis during cell cycle progression, whereas WEE2 governs cell cycle progression during meiosis. Recent studies have identified WEE1 as a potential therapeutic target in several cancers, including therapy-resistant triple-negative breast cancer. Adavosertib’s clinical promise was challenged by inter-individual variations in response and side effects. Because of these promising preclinical outcomes, other WEE1 kinase inhibitors (Azenosertib, SC0191, IMP7068, PD0407824, PD0166285, WEE1-IN-5, Zedoresertib, WEE1-IN-8, and ATRN-1051) are being developed, with several currently being evaluated in clinical trials or as an adjuvant to chemotherapies. Preclinical studies show WEE1 inhibitors induce MHC class 1 antigens and STING when given as combination therapies, suggesting potential additional therapeutic opportunities. Reliable predictors of clinical responses based on mechanistic insights remain an important unmet need. Herein, we review the role of WEE1 inhibition therapy in breast cancer.

## 1. Introduction

WEE1 kinase was first identified as a thermo-sensitive regulator of cell size and division in yeast. Its tyrosine kinase activity was shown to regulate the cell cycle G2/M checkpoint through inhibitory phosphorylation of CDK1 and CDK2, and preventing premature mitotic entry, especially in the presence of DNA damage. This ensures that DNA replication and repair are completed prior to mitosis, thereby maintaining genomic stability [1].

The WEE kinase family consists of three proteins: WEE1, PKMYT1, and WEE2 (WEE11B) [2]. WEE1 exclusively mediates phosphorylation at the Tyr15 residue of CDK1 kinase, whereas PKMYT1 is dual-specific for Tyr15 and Thr14 residues [3]. WEE1 and PKMYT1 regulate entry into mitosis during cell cycle progression, while WEE2 regulates cell cycle progression during meiosis [2,4,5]. Under physiological circumstances, the cell cycle entry into the G2 phase is governed by the CDK1/Cyclin B complex, also known as the mitotic-promoting factor (MPF) [3,6,7]. The activity of MPF is under the control of WEE1 family kinases and Cdc25 phosphatases. WEE1 phosphorylates the Tyr15 residue of CDK1 kinase [3], which inhibits the complex, thereby restricting mitotic entry in the presence of DNA damage [1], as seen in Figure 1 [8]. In the absence of DNA damage, this inhibitory phosphorylation is removed by CDC25C phosphatases [3,9,10].

DNA damage occurring during replication is repaired, primarily at the G1 and G2 phases of the cell cycle. DNA repair at the G1/S checkpoint is governed by the tumor suppressor gene p53. This gene is mutated in almost half of all human malignancies [11]. Without functional p53 protein, cancer cells progress past the G1/S checkpoint and predominantly depend upon the G2/M checkpoint to prevent excessive DNA damage, which is under the control of WEE1 [12,13,14,15]. DNA damage activates ATR/Chk1, and Chk1 phosphorylates and activates Wee1, which then subsequently phosphorylates and inhibits CDK1/Cyclin B function, ultimately resulting in G2 phase arrest, potentially allowing for DNA repair [16,17]. In cancer cells, WEE1 kinase inhibition abrogates DNA repair at a crucial checkpoint prior to mitosis. This results in early mitotic entry, leading to mitotic catastrophe via several intra-mitotic mechanisms such as centromere fragmentation [18], genetic damage, and, ultimately, apoptotic cell death [19,20].

In addition to WEE1 regulation at the G2/M checkpoint, checkpoint regulation occurs within the S phase (intra-S phase) [21]. Pre-replication complexes allow cells that have reached the S phase to initiate replication, preceded by the activation of CDK2, which is regulated by Tyr15 phosphorylation, similar to CDK1. This regulation is balanced by WEE1 and CDC25A [22]. WEE1 kinase, therefore, also regulates CDK1/CDK2 activity through the inhibitory phosphorylation of kinases to control entry into mitosis and DNA replication during the S phase as well as the G2/M checkpoint (Figure 1) [23].

Recent reports show that WEE1 inhibition induces replication stress by CDK1/2-dependent aberrant firing of replication origins and reduced replication fork processivity, which leads to subsequent nucleotide shortage [24,25]. WEE1 prevents DNA damage and chromosome pulverization through indirect inhibition of MUS81. MUS81 is an endonuclease responsible for the formation of heterodimeric complexes that recover stalled replication forks during prolonged S phase arrest as well as resetting DNA junctions between twin chromatids during homologous recombination (HR). Indeed, the lack of MUS81 endonuclease regulation by WEE1 may lead to excessive cleavage of unwanted DNA structures, an abundance of replication forks, and a delayed replication progression, ultimately leading to increased genomic instability [26,27,28].

**Figure 1 ijms-26-05701-f001:**
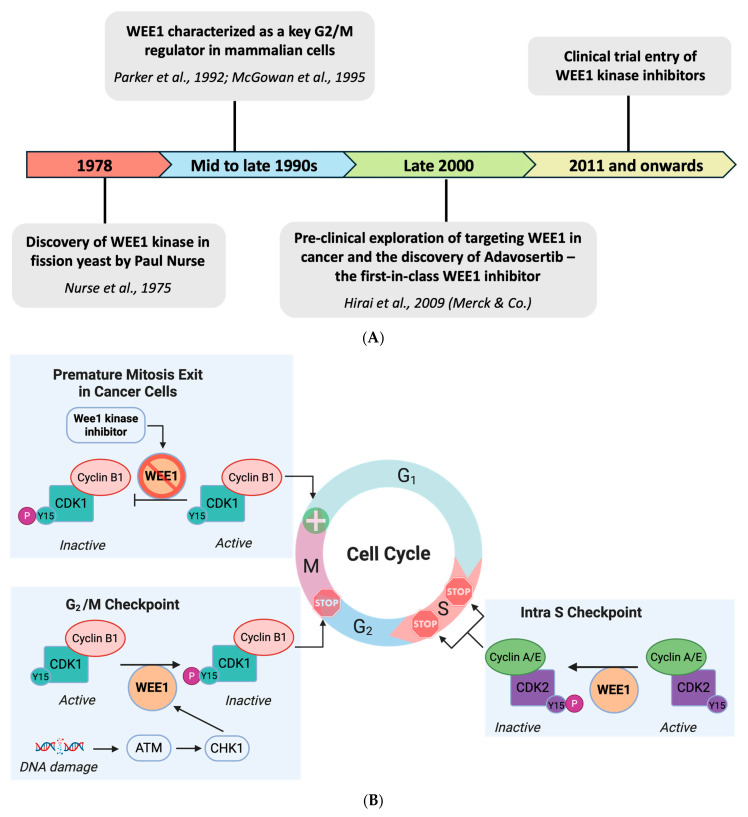
(**A**) Graphical timeline summarizing the evolution of WEE1 kinase as a therapeutic target in cancer [29,30,31,32]. (**B**) Schematic representation of the role of WEE1 in regulating intra-S checkpoint, G2/M checkpoint, and mitosis exit.

## 2. WEE1 Kinase in Breast Cancer

Breast cancer affects approximately 12% of women worldwide and contributes to 14% of all cancer deaths [33,34,35]. Based on the expression status of the hormone receptors estrogen receptor (ER), progesterone receptor (PR), and the human epidermal growth factor receptor (HER2), breast cancer can be categorized into Luminal A, Luminal B, HER2-overexpressing/enriched, triple-negative/basal-like breast cancer (TNBC), and normal-like breast cancer (NLBC) [36,37,38,39,40,41,42,43] (Figure 2, Table 1).

Targeted therapies are the current standard of care for receptor-expressing subtypes. ERα inhibitors (like Tamoxifen and Raloxifene) and HER2 inhibitors (like Trastuzumab and Pertuzumab) are administered to Luminal A/B and HER2+ diagnosed breast cancer patients, respectively [44,45]. TNBC is the most aggressive subtype, and the lack of expression of these receptors limits its targeted therapy options [46]. Current FDA-approved treatments include chemotherapy using Anthracyclines (Doxorubicin and Epirubicin) and Taxanes (Docetaxel and Paclitaxel), but the development of cardiotoxicity and chemotherapy resistance are two concerning side effects [47]. The immune checkpoint inhibitor pembrolizumab was approved by the FDA for early-stage TNBC based on the results of a clinical trial (KEYNOTE-522) demonstrating improved pathological complete response (pCR) and event-free survival (EFS) rates in patients receiving pembrolizumab compared to those receiving a placebo [48]. In the metastatic pretreated setting, pembrolizumab did not show a survival advantage over chemotherapy (KEYNOTE-119 (ClinicalTrials.gov, NCT02555657)) [49]. In previously untreated patients with higher PDL1 levels, the addition of pembrolizumab to chemotherapy resulted in significantly longer overall survival than chemotherapy alone [50], suggesting therapies that induce PDL1 may be useful adjuncts. The DNA damage response in TNBC is also being targeted using poly ADP ribose polymerase (PARP) inhibitors, including Olaparib and Talazoparib [51]. However, as PARP inhibitors are mainly effective in patients with *BRCA* mutations, and their use is associated with serious and frequent side effects, their clinical utility remains limited [47]. Promising results with Sacituzumab govitecan, an antibody–drug conjugate (ADC) composed of an antibody targeting the human trophoblast cell-surface antigen 2 (Trop-2) coupled to a topoisomerase I inhibitor (SN-38), led to FDA approval for mTNBC patients who have received two or more prior systemic therapies. Sacituzumab govitecan showed improved progression-free and overall survival when compared with single-agent chemotherapy among patients with metastatic triple-negative breast cancer, although grade 3 or higher side effects (myelosuppression (~50%) and diarrhea (10%)) were frequent. A similar prevalence of adverse events was confirmed in a recent real-world study [52]. Although there were no complete responses amongst the 149 patients, progression-free survival was 5.7 months [52]. Interestingly, treatment with a humanized monoclonal antibody to CCR5 (leronlimab) used in a pooled analysis of pretreated metastatic TNBC patients showed no grade three toxicities related to therapy and a 3 year survival of 19.8% [53,54]. Overall, 88% of patients who received a dose of 525 or 700 mg showed an upregulation of PD-L1 in circulating tumor cells. Patients who showed a significant induction of PD-L1 on their CTC with leronlimab, and received an immune check point inhibitor, were alive (>48 months) [54]. In summary, there is a need to identify novel therapeutic approaches to TNBC treatment, and WEE1 kinase has emerged as a candidate.

**Table 1 ijms-26-05701-t001:** Summary of characteristics, prevalence, and treatment approaches for different subtypes of breast cancers [44,55,56,57,58].

Subtype	Receptor Status	Characteristics	Prevalence (%)	Treatment	Five-Year Relative Survival Rate
Luminal A (Lum A)	ER+ PR+ (and/or) HER2−	Hormone receptor+ Low Ki-67 Better prognosis	60–70%	Hormonal therapy (Tamoxifen, Aromatase inhibitors)	~94.4%
Luminal B (Lum B)	ER+ PR+ HER2+/−	Higher Ki-67 More aggressive than Luminal A. Often associated with *BRCA2* mutation	60–70%	Hormonal therapy, targeted therapy	~90.7%
HER2-overexpressing (HER2+)	HER2+ ER− PR−	*HER2* gene amplification Aggressive but responsive to HER2-targeted therapy. Often linked to p53 mutation	10–15%	Targeted therapy (HER2 inhibitors: Trastuzumab, Pertuzumab)	~84.8%
Triple Negative Breast Cancer (TNBC)	ER− PR− HER2−	Lacks ER, PR, and HER2 expression; highly aggressive Poor prognosis Often linked to *p53, BRCA1* mutations	15–20%	Chemotherapy, surgery, radiation, PDL1 and PARP inhibitors	~77.1%
Normal-like Breast Cancer (NLBC)	Variable	Share molecular features with normal breast tissue	N/A	Chemotherapy, surgery, radiation	N/A

The specific role of WEE1 in breast cancer is incompletely understood. While a study reported low WEE1 expression in breast tumors, which was independent of tumor grade in comparison to normal tissue pathology based on the analysis of the data in in the Oncomine cancer microarray database [59], another study with 8636 primary breast cancers, including 1847 TNBC, gathered from 36 public gene expression data sets [60] showed that in TNBC, high-level *WEE1* gene expression was associated with poor prognosis [61]. Our overall survival analysis based on the TCGA BRCA data set (http://gepia.cancer-pku.cn, accessed on 20 May 2025), which included 213 patients, showed that with stringent high (90%) and low (10%) expression cutoffs, *WEE1*-overexpressed breast cancer patients have a significantly lower overall survival (Figure 3). Moreover several other studies have shown that high levels of WEE1 expression are associated with worse prognostic factors, including metastasis, increased proliferative biomarkers, and resistance to treatment, in other cancers including glioblastoma, glioma, gastric cancer, and malignant melanoma [20,62,63,64]. Interestingly, RNAi-based functional genomic screening of the human tyrosine kinome identified *WEE1* as a potential target kinase [65]. These contrasting roles of WEE1 are also seen in biological experimentation. One study has shown that *WEE1* deletion in mammary epithelial cells of mice leads to increased tumor growth and concluded that WEE1 functions as a tumor suppressor [59]. However, in tumors that rely on G2/M checkpoint arrest for DNA damage repair, preclinical and clinical studies that explored WEE1 inhibition showed a reduction in the growth of a wide variety of cancers, including breast cancer.

## 3. WEE1 Kinase Inhibitors for Breast Cancer Treatment

WEE1 is expressed at high levels in several cancer types, including breast cancer (Figure 4A–C), hepatocellular carcinoma, leukemia, melanoma, and adult and pediatric brain tumors [66,67,68,69,70]. The major consequences of WEE1 inhibition include the accumulation of DNA damage, alterations in cell cycle regulation, and induction of apoptosis [71].

WEE1 inhibition has been observed upon the use of bioflavonoids, notably quercetin, a bitter-tasting molecule found in plants/seeds such as capers and buckwheat [72]. It was also found that Kava Chalcone (specifically Flavokawain A), a molecule contained in kava extract, preferentially inhibits the growth of HER2+ breast cancer cells via the downregulation of Cdc2 inhibitors, including WEE1 kinase [73]. The IC_50_ of Flavokawain A on the growth of HER2-overexpressing SKBR3 and MCF-7/HER2 cells is 10 and 13.6 μM, respectively, versus 38.4 and 45 μM for MCF7 and MDA-MB-468 cells, respectively [73]. Flavokawain A at a concentration of 4 μM inhibits the colony formation of MCF/HER2 and MCF7 by 80% and 54%, respectively [73]. Nevertheless, biopharmaceutical-developed inhibitors are superior due to better control over pharmacokinetics, bioavailability, and chemical standardization.

Adavosertib (AZD-1775, formerly MK-1775) is the most widely studied WEE1 kinase inhibitor, extensively evaluated in both preclinical and clinical settings for its potential in breast cancer treatment following a licensing agreement between Merck and AstraZeneca. In an in vivo study, 6-week-old female nude mice (*n* = 10 per group) had their mammary fat pads orthotopically injected with HER2-positive breast cancer cell lines, including BT474R, HCC1954, and T47D. The tumor-bearing mice were treated with Adavosertib (120 mg/kg, five days on, two days off) or vehicle control. Adavosertib administration effectively reduced tumor growth and tumor burden in BT474R and T47D cells [74].

## 4. Combination Strategies to Enhance the Efficacy of WEE1 Kinase Inhibitors

Preclinical studies have demonstrated that WEE1 inhibition enhances the cytotoxic effects of several chemotherapies, leading to increased tumor regression and reduced therapy resistance (Figure 5). This section of the review highlights the key preclinical studies evaluating WEE1-inhibitor-based combination therapies in breast cancer models.

### 4.1. Combination Therapy with CDK4/6 Inhibitors

Cyclin D1 encodes the regulatory subunit of a holoenzyme that includes the CDK4 or CDK6 subunit, which promotes normal cell cycle G1/S transition. The *CCND1* gene is amplified in ~10–20% of primary breast cancers and preferentially occurs in ERα^+^ tumors [75]. Cyclin D1 protein overexpression in *ERα*-positive breast cancers correlates with poor responses to endocrine agents. Genetic aberrations of the cyclin D1–CDK4/6 pathway are linked with poor clinical outcomes in ERα^+^ breast cancer [70]. Mechanistically, this may be because cyclin D1 determines a cascade of estrogen-dependent gene expression and enhances homology-directed DNA repair [76]. The IND enabling experiments, in which cyclin D1 anti-sense inactivation reduced breast tumor growth [77], led to the rationale development of therapies targeting the cyclin D1 holoenzyme [78].

The CDK inhibitor Palbociclib (PD0332991) is a well-tolerated, highly specific inhibitor of CDK4 (IC_50_ = 11 nM) and CDK6 (IC_50_ = 16 nM) that suppresses the growth of estrogen-receptor-positive breast cancer [79,80,81]. However, patients relapse with acquired resistance to Palbociclib when combined with endocrine treatment through poorly characterized mechanisms [78]. Resistance to Palbociclib can occur due to ERBB signaling, CDK7, and G2/M–checkpoint proteins such as WEE1 [82]. Pancholi et al. [82] showed that inhibition of WEE1 using Adavosertib effectively suppressed proliferation in some breast cancer cell lines tested (MCF7, T47D, HCC1428, ZR75.1, and SUM44) and enhanced the sensitivity of Palbociclib-resistant cells compared with parental cells. Adavosertib inhibited the growth of a Palbociclib-resistant PDX model of metastatic Erα^+^  CCND1-driven breast cancer by 70% after 60 days of treatment in Swiss nude mice in vivo.

Dinaciclib, an inhibitor of CDK1/2/5/9, conveyed functional synergy with Adavosertib. Intraperitoneal injection of Dinaciclib (25 mg/kg) followed by oral administration of Adavosertib (50 mg/kg) showed significant synergistic effects in Cyclin-E low TNBC cell lines, including SUM149, SUM159, and MDA231 in vitro and in vivo models. Similar results were also observed in patient-derived xenograft (PDX) models. Compared to Dinaciclib monotherapy, the sequential combination treatment with Adavosertib reduced the tumor volume by 60% in PDX models [83].

Studies in ERα^+^ endocrine-resistant and CDK4/6-inhibitor-resistant MCF7 and T47D breast cancer cells, targeting the G2/M checkpoint, showed that Adavosertib significantly decreased cell proliferation and increased G2/M arrest, apoptosis, and γ-H2AX levels (a marker for DNA double-stranded breaks) in resistant cells compared with sensitive cells. However, these antiproliferative effects were compromised when combined with CDK4/6 inhibitors like Ribociclib. This study highlights the importance of identifying molecular signatures and resistance mechanisms, as the response to combination therapy can vary depending on specific tumor characteristics [84].

### 4.2. Combination Therapy with ATR/CHK1 Inhibitors

Ataxia telangiectasia and Rad3-related (also known as ATR) is a serine/threonine kinase member of the PI3K family that plays a role in breast cancer by activating in response to single-stranded DNA strands, which only occur as intermediates during DNA repair, most notably at stalled replication forks undergoing nucleotide excision repair (NER) or homologous recombination repair (HR). Once activated, ATR phosphorylates CHK1, another serine/threonine kinase that facilitates the DDR response at the S phase, G2/M phase, and M phase, serving as a signal transducer and a regulator of late-origin firing, elongation, and monitoring replication fork integrity [85,86]. CHK1 also phosphorylates WEE1 kinase to initiate cell cycle arrest [86].

Studies conducted in breast cancer cells have shown that ATR inhibition provides additional genotoxic stress. Jin et al. 2018 utilized AZD-6738, an ATR inhibitor, in conjunction with Adavosertib to produce a synergistic effect on mitotic instability, noting a 2.5-fold increase in replication fork stall time compared to Adavosertib alone [87]. The combination of the CHK1 inhibitor, SRA737, with Adavosertib enhanced tumor growth inhibition in the *BRCA1*-mutated breast cancer cell line MDA-MB-436 xenograft model [88].

### 4.3. Combination Therapy with PARP Inhibitors

Preclinical studies have shown that Adavosertib may broaden the utility of the PARP inhibitor Olaparib in treating TNBC [89]. Cancer cells can escape T cell recognition by reducing MHC expression levels [90]. Combining PARP and WEE1 inhibition upregulated MHC class I molecule expression in *BRCA1/2* wildtype TNBC cell lines [91]. Teo et al. also showed that combining Olaparib with Adavosertib reduced tumor growth by increasing the anti-tumor immune responses and activating the STING pathway in *BRCA1/2* wildtype TNBC [91].

### 4.4. Combination Therapy with Platinum Containing Compounds

Adavosertib enhanced tumor cell killing by carboplatin, allowing a reduction in the dose and thereby a reduction in dose-dependent toxicities of carboplatin [92]. Cisplatin is another drug that belongs to the same drug family. A combination of Adavosertib and cisplatin exhibited enhanced sensitivities in *BRCA*-proficient TNBC cells [89].

### 4.5. Combination Therapy with HER2-Targeted Therapies

Cancer is a genetically, epigenetically, and phenotypically heterogeneous disease [93]. A subpopulation of cells exists within a malignant cluster of cells called the cancer stem cells (CSCs), which account for around 0.1–10% of the tumor population [94,95]. These CSC populations are believed to be crucial in tumor initiation and progression and resist conventional therapies [96,97]. These CSCs, which escape the therapeutic measures, are responsible for tumor regrowth and relapse owing to their self-renewal and differentiation abilities [95]. Recently, many studies have been going on to develop therapeutic strategies to target these CSCs directly [98,99,100]. One of the standard practices for treating HER2-positive breast cancer is Trastuzumab administration, which is a humanized monoclonal antibody that blocks HER2 activation [101]. MUC1 has been previously recognized as a biomarker for Trastuzumab resistance, and in a separate study, MUC1 upregulation has been associated with CSC growth [74,102,103]. In 2020, Sand et al. successfully showed that Adavosertib effectively targeted cancer stem-like properties by suppressing MUC1 expression levels, thereby increasing the sensitivity of Trastuzumab-resistant cells [74]. Other studies also suggest that overexpression of checkpoint inhibitors like WEE1 could be the reason behind the resistance of CSCs to DNA-damaging treatments [104]. These findings raise the prospect of WEE1 kinase inhibition to target CSCs precisely.

### 4.6. Combination Therapy with Apoptosis Inducers

The Tumor Necrosis Factor (TNF)-Related Apoptosis-Inducing Ligand (TRAIL) induces apoptosis in cancer cells [105]. Pre-incubation with the WEE1 inhibitor synergized with TRAIL-mediated apoptosis. Basal B/TNBC cell lines (MB231, HCC38, MB157, BT549, and Hs578T) showed more sensitivity in comparison to basal A/TNBC (BT20 and HCC1937) and HER-2-amplified (BT474 and SKBR3) cell lines. However, the ERα^+^ cell lines (MCF7 and T47D) were relatively resistant to mono- and combination therapy. Enhanced activation of caspases, especially Caspase-8 and Caspase-9, which activate both extrinsic and intrinsic apoptotic pathways, respectively, was reported to be the underlying mechanism of the synergistic action of WEE1 inhibitors and TRAIL [106].

### 4.7. Combination Therapy with Antimetabolites

Antimetabolites are a class of chemotherapeutic drugs that interfere with cell division and prevent the replication of cancer cells [107]. Numerous studies have explored the combination of WEE1 inhibitors with antimetabolites, like gemcitabine, in cancers, including pancreatic cancer [108]. Pitts et al. investigated the efficacy of Adavosertib in conjunction with the antimetabolite Capecitabine in TNBC PDX models and TNBC cell lines. In the combination regimen, the reduction in tumor volume ranged from 50 to 70% in different TNBC PDX models compared to Capecitabine monotherapy. Enhanced apoptosis, increased DNA damage response (based on γ-H2AX levels), and cell cycle arrest were observed in the combination regimen of these two drugs [109].

## 5. Resistance to WEE1 Kinase Inhibitors and Predictors of Response

While WEE1 kinase inhibitors have shown promising therapeutic potential in breast cancer, therapy resistance occurs. Alterations in DNA damage repair pathways, cell cycle checkpoints, and apoptosis evasion mechanisms become activated. The identification of reliable biomarkers for likely responders to WEE1 kinase inhibitors is essential for patient therapeutic stratification (Figure 6).

### 5.1. PKMYT1 Upregulation

PKMYT1 is another member of the WEE family kinases and in certain circumstances may be functionally redundant with WEE1 kinase. While WEE1 inhibits CDK1/2 in the nucleus, PKMYT1 is located in the Golgi apparatus and targets CDK1/2 in the cytoplasm, indicating distinct functions [2]. PKMYT1 may undertake the role of WEE1 when inhibited by specific compounds and become a prospective mechanism by which resistance to Adavosertib is achieved. Lewis et al. observed this phenomenon wherein PKMYT1 upregulation conferred resistance to Adavosertib in TNBC cell lines [110].

### 5.2. Cyclin E Overexpression

Cyclin E overexpression has been implicated as a prognostic marker for breast cancer, most notably TNBC. Cyclin E activates Cdk2 and is most prevalent in the G1/S phase. Chen et al. found a higher prevalence of Cyclin E mutations in TNBC (52% in The Cancer Genome Atlas and 40% in the METABRIC database) compared to ERα^+^ breast cancer (3% in TCGA and 2.3% in METABRIC) [83]. A total of 77% of patients with recurring TNBC have high expression of Cyclin E. These Cyclin-E-overexpressing cells rely on WEE1 kinase to reduce replicative stress and allow for DNA damage repair. Additionally, it was found that Cyclin E overexpression is accelerated by Cdk2-dependent activation of DNA replication stress pathways, placing even more burden on WEE1 kinase to sustain checkpoint inhibition for DNA damage repair. Therefore, Cyclin E overexpression may be a biomarker to identify TNBC patients who may respond well to Adavosertib monotherapy.

### 5.3. BRCA Mutations

Breast cancer patients harboring mutations in *BRCA1/2* genes are known to have the worst outcomes. Studies conducted with in vivo and in vitro model systems have shown that *BRCA* mutations can lead to high Cyclin E1 expression. *BRCA-1*-mediated stabilization of Cyclin E1 is achieved by reducing the phosphorylation at T62 residue [111,112,113]. Hence, *BRCA* mutations may have the potential to be used as a predictive biomarker for WEE1 inhibitor response in the future.

### 5.4. ATRX Deficiency

ATRX (Alpha Thalassemia/Mental Retardation Syndrome X-Linked) is a chromatin remodeling protein involved in maintaining genomic stability and regulating gene expression [114,115]. *ATRX*-deficient cells exhibit increased sensitivity to WEE1 inhibition, as the disruption of WEE1 exacerbates replication stress, leading to mitotic catastrophe and cell death [87,116]. *ATRX* status is being evaluated in clinical trials for breast carcinoma. These Phase 1 clinical studies aim to assess the potential of ATRX as a biomarker and its implications for targeted therapies [117].

### 5.5. EZH2 Deficiency and STING Pathway Activation

EZH2 (enhancer of zeste homolog 2) is a histone methyltransferase. It mediates trimethylation of H3K27 and plays a role in gene silencing. EZH2 is a strong inhibitor of anti-tumor immunity and responsiveness to checkpoint inhibitors. Research has shown that inhibition of EZH2 activates a double-stranded RNA (dsRNA)-STING-interferon stress axis, resulting in increased response to PD-1 checkpoint blockade in prostate cancer. *EZH2*-deficient tumors may, therefore, be more receptive to therapies targeting the DNA damage response, including WEE1 inhibitors [118].

### 5.6. H3K36Me3 Deficiency

Histone H3K36 trimethylation (H3K36me3) is reported to be frequently lost in many types of cancers. H3K36me3-deficient cancers are hypersensitive to WEE1 inhibition. This synthetic lethal interaction is reported to be mediated by RRM2, a subunit of ribonucleotide reductase. H3K36me3 facilitates the transcription of RRM2, which governs nucleotide synthesis and DNA replication. WEE1 inhibition in H3K36me3-deficient cells rescues RRM2 expression, leading to dNTP depletion, S-phase arrest, and apoptosis. Treatment of H3K36me3-deficient cell tumors with WEE1 inhibitors showed increased cell killing [119]. Adavosertib also inhibited H3K36me3-deficient tumor xenografts and sensitized cells to immunotherapy [119,120,121].

### 5.7. LKB1 Deficiency

LKB1 (liver kinase B1) is constitutively active in cells [122], serving as a tumor suppressor that regulates cellular metabolism and energy homeostasis. LKB1 is phosphorylated and inactivated by cyclin D1-Cdk4/Cdk6 [123], and LKB1 phosphorylation by oncogenic B-RAF compromises the ability of LKB1 to bind and activate AMPK [124]. Low expression of LKB1 correlates with markers of unfavorable breast cancer prognosis, including increased E-cadherin and HMW-CK expression. Deficiency of LKB1 promotes metastasis in breast cancer cells [125]. *LKB1*-deficient cancer cells are vulnerable to WEE1 inhibition, as the added stress overwhelms the cells’ capacity to maintain genomic integrity and hence leading to cell death. In vitro, *LKB1* deficiency enhanced DNA damage and apoptosis in response to Adavosertib exposure compared with wildtype *LKB1* cells [126].

### 5.8. SETD2 Deficiency

SETD2 [119] (the methyltransferase SET domain-containing 2) deposits the H3K36me3 epigenetic marker and is crucial for DNA repair and transcriptional regulation. Loss of SETD2 function leads to defective homologous recombination repair. *SETD2*-deficient cells are hypersensitive to WEE1 inhibitors, as this inhibition further compromises DNA repair, resulting in increased genomic instability and cell death [127]. WEE1 inhibition targets *SETD2*-deficient cells via S-phase arrest and is distinct from p53-deficient cells [128]. Shen et al. have shown that circular RNA circ_*SETD2* represses breast cancer progression by modulating the miR-155-5p/SCUBE2 axis [129]. The discovery of *SETD2* as a frequently mutated gene in phyllodes tumors of the breast (PT) suggests SETD2 may serve as a biomarker for this aggressive breast cancer subtype [130].

### 5.9. CDK2 Expression

Deletion of *CDK2* desensitizes cancer cells to WEE1 inhibition. Deficiencies in CDK2 can lead to resistance against the cytotoxic effects of WEE1 inhibitors. This resistance is primarily due to a reduction in DNA damage during the S phase. Mutations in SKP2 and CUL1 also confer WEE1 inhibition resistance. Elevated CDK2 activity resulting from WEE1 inhibition causes DNA damage during the S phase. However, inhibiting CDK2 can prevent this DNA damage and the subsequent bypassing of the G2 phase, although it does not rectify defects in cytokinesis [131].

### 5.10. p53 Deficiency/Mutations

Many preclinical and clinical trials showed promising results when the WEE1 kinase inhibitor Adavosertib was used to treat cancers with p53 mutations [132]. A Phase II clinical trial conducted on women with *TP53*-mutated platinum-sensitive ovarian cancer (NCT01357161) showed that the addition of Adavosertib to chemotherapy significantly improved the progression-free survival in patients [132]. However, *TP53* mutational status is not a reliable standalone predictive biomarker for WEE1 inhibitors’ response [133]. A review analyzing ongoing clinical trials of WEE1 inhibitors in various cancers reported conflicting evidence on the predictive value of p53 mutations [133]. However, when p53 is mutated, Chen et al. [83] and Fallah et al. [84] observed resistance to Adavosertib in some, but not all, cell lines.

### 5.11. PTEN Loss

PTEN (Phosphatase and Tensin Homolog) functions as a tumor suppressor via its action as a phosphatase to inhibit the PI3K/Akt signaling pathway, which promotes progression through the G1 and G2 checkpoints. Alterations in PTEN are frequently observed in cancers [134]. *PTEN*-deficient breast cancers have worse disease-free and overall survival rates [135] and have been linked to reduced efficacy of CDK4/6 inhibitors and PI3K/AKT/mTOR pathway inhibitors [136]. *PTEN*-deficient cancer cells often exhibit increased sensitivity to WEE1 inhibitors. Enhanced cell death is due to the combined disruption of PI3K/AKT signaling and WEE1-mediated cell cycle controls. Brunner et al. found a significant association between low PTEN protein expression in TNBC cell lines and increased sensitivity to Adavosertib compared to TNBC cell lines that recovered from Adavosertib monotherapy [137].

### 5.12. c-Jun Loss

c-Jun protein is an essential component of the activator protein 1 (AP-1) transcription factor complex and is overexpressed in most tumors, including breast cancer [138,139]. Deletion of *c-Jun* reduces cell migration and invasion through inhibition of c-Src and hyperactivation of ROCK II kinase [140]. In mammary epithelial tumor cells, c-Jun enhances proliferation, invasiveness, and stemness [138]. The activity of c-Jun is regulated by post-translational modifications controlled by mitogen-activated protein kinase (MAPK) family kinases, including c-Jun N-terminal kinase (JNK), extracellular-signal-regulated kinase (ERK), and p38 kinase [141]. In MCF-7 breast cancer cells, the antiproliferative effects of Tamoxifen were reversed by c-Jun overexpression through activating the protein kinase C (PKC) pathway [142]. In MDA-MB-231 and MCF-7 cell lines, increased expression of the RNA-binding protein Tristetraprolin (TTP) induced cell cycle arrest by targeting c-Jun, a key component of the AP-1 transcription factor, through inhibition of the NFkB signal pathway [139]. Suppression of c-Jun expression leads to increased WEE1 expression, which suppresses cell proliferation. Overexpression of c-Jun into TTP-expressing cells reduced WEE1 expression and restored cell proliferation [139]. c-Jun negatively regulates WEE1 expression.

## 6. Clinical Trials Targeting WEE1 Kinase in Breast Cancer

As preclinical studies have demonstrated the efficacy of WEE1 kinase inhibitors in breast cancer, clinical evaluation of WEE1 inhibitors has begun.

A Phase Ib study assessed the safety, tolerability, and efficacy of Adavosertib treatment in patients diagnosed with advanced solid tumors (NCT02482311). The Part B cohort of this study included 80 patients who were diagnosed with ovarian cancer (46 patients), small-cell lung cancer (15 patients), and TNBC (19 patients) who had received prior regimens. The TNBC patients were also further divided into *CCNE1/MYC/MYCL1/MYCN* biomarker amplified (6 patients) and biomarker non-amplified (13 patients) tumor subgroups. The overall disease control rate (DCR) was moderate: 50% for the biomarker-amplified cohort and 69% for the biomarker non-amplified cohort. Even though the patients with stable disease showed varying durations, the biomarker non-amplified TNBC cohort exhibited a higher percentage of stable disease. Progressive disease was more prevalent in the biomarker-amplified cohort (50%). A relatively low median progression-free survival of 2 and 3.1 months was observed in the biomarker-amplified and biomarker-non-amplified TNBC subgroups, respectively [143].

A Phase II trial of Adavosertib was conducted in 18 patients with *SETD2*-altered advanced solid tumor malignancies (NCT03284385). Unfortunately, although some patients experienced prolonged stable disease, no objective responses could be noticed. Predictive biomarkers were not explored in this study. More than 25% of the patients experienced treatment-emergent adverse effects like nausea, anemia, diarrhea, and neutropenia. This study also suggests that a combination regimen may yield a better tumor response [144].

Another multicenter Phase II trial was focused on assessing the clinical utility of Adavosertib as a monotherapy in patients with refractory solid tumors harboring *CCNE1* amplification (NCT03253679). The study reported a manageable toxicity level as well as promising clinical activity with an objective response rate (ORR) of 27%, particularly in epithelial ovarian cancer. Common treatment-associated toxicities included gastrointestinal and hematologic events. Additionally, a baseline molecular profiling of the 30 patients enrolled in the study was conducted to explore potential biomarkers for predictive response. In 90% of the patients, *CCNE1* amplification was followed by a concurrent aberration in the *TP53* gene. *AKT2* amplification (23%), *MYC* amplification (17%), *CCND2* amplification (10%), and *NOTCH1* mutations (10%) were the other prominent genomic aberrations observed [145]. Specific data on breast cancer patients were not detailed in this study, but these findings suggest potential applicability in *CCNE1*-amplified breast cancers.

The clinical utility of Adavosertib was also assessed in combination with other existing chemotherapeutic agents (Table 2). A Phase II study (NCT03012477) combining Adavosertib with cisplatin in metastatic triple-negative breast cancer showed an objective response rate (ORR) of 26% and a median progression-free survival (PFS) of 4.9 months. However, adverse events occurred in over 20% of patients. Another Phase I study (NCT02617277) tested Adavosertib with the CDK inhibitor durvalumab in advanced solid tumors, revealing limited anti-tumor activity but notable toxicities such as fatigue, diarrhea, and anemia. A broader Phase I study (NCT00648648) evaluated Adavosertib alone and in combination with cisplatin, carboplatin, or gemcitabine, establishing tolerable doses that exceeded pharmacokinetic thresholds for efficacy. Across all studies, common treatment-related adverse events included gastrointestinal disturbances, fatigue, and hematologic toxicities like thrombocytopenia and neutropenia.

Despite compelling scientific rationale and promising preclinical data, the expected clinical utility of Adavosertib was not achieved in the breast cancer clinical studies. Several oncology-focused biopharmaceutical companies are actively developing more selective and tolerable WEE1 inhibitors, details of which are explained in Table 3. Additional WEE1 kinase inhibitors have been developed and their IC_50_ for WEE1 kinase inhibition were 3.9 nM for Azenosertib (ZN-c3), 22.3 nM for SC0191, 2.25 nM for ATRN-1051 (APR-1051), 0.524 μM for IMP7068 (WEE1-IN-10, Potrasertib), 97 nM for PD0407824 (47 nM for Chk1 inhibition), 24 nM for PD0166285, 0.8 nM for WEE1-IN-5, 0.8 nM for Debio 0123 (Zedoresertib) and 0.98 nM for WEE1-IN-8. Some of these inhibitors, like PD0166285 and PD0407824, had limited clinical utility due to issues with specificity and potency levels. The clinical trials for other WEE1 kinase inhibitors including Azenosertib (NCT04814108, NCT04972422, and NCT04158336), SC0191 (NCT06363552), ATRN-1051 (NCT06260514), IMP7068 (NCT04768868), Debio 0123 (NCT03968653 and NCT04855656) are in clinical trials. Azenosertib and SC0191 demonstrated impressive anti-tumor efficacy in studies conducted in other solid tumors. In 2024, Aprea Therapeutics initiated a Phase 1 trial to evaluate the safety and tolerability of the novel WEE1 inhibitor APR-1051 in patients with advanced solid tumors (ACESOT-1051 and NCT06260514). Preliminary results from this study indicate that this next-generation WEE1 inhibitor is safe and well tolerated. ACESOT-1051 is still an ongoing first-in-human Phase 1 study, with a special focus on cancer-associated gene alterations, such as overexpression of *CCNE1/2*, loss-of-function mutation in *FBXW7*, *PPP2R1A*, or *KRAS GLY12* or *GLY13* with *TP53* co-mutation [146].

**Table 2 ijms-26-05701-t002:** Information on clinical studies using Adavosertib as part of a combination treatment regimen [147,148,149].

Clinicaltrials.gov	Clinical Study Title	Adavosertib Combination Regimen	Dose Information	Results/Adverse Effects
NCT03012477	A Phase II Study of Cisplatin + Adavosertib in Metastatic Triple-negative Breast Cancer and Evaluation of pCDK1 as a Biomarker of Target Response	Combination with cisplatin	Cisplatin 75 mg/m^2^ IV followed 21 days later by cisplatin plus Adavosertib 200 mg oral twice daily for 5 doses every 21 days	The objective response rate (ORR) was 26% and fell below the pre-decided cutoff of 30% ORR. The median progression-free survival (PFS) was 4.9 months. Treatment-related adverse events were observed in >20% of patients.
NCT02617277	Open-label, multi-center, phase I study to assess safety and tolerability of Adavosertib plus durvalumab in patients with advanced solid tumors	Combination with durvalumab	RP2D 150 mg bd (3 days on, 4 days off; treatment D15–17, D22–24) + durvalumab 1500 mg (D1 q28d)	Preliminary evidence of limited anti-tumor activity of Adavosertib + durvalumab. The most frequent (in >5% of patients) treatment-emergent grade ≥ 3 toxicities were Fatigue; Diarrhea; Nausea; Anemia; Abdominal pain.
NCT00648648	Phase I study evaluating wee1 inhibitor Adavosertib as monotherapy and in combination with gemcitabine, cisplatin, or carboplatin in patients with advanced solid tumors	Part 1: Monotherapy Part 2: Combination with: Carboplatin, Cisplatin or Gemcitabine	Single doses of: 325 mg, 650 mg or 1300 mg 225 mg bd for 2.5 days week 1 + carboplatin AUC 5 D1 q21 200 mg bd for 2.5 days week 1 + cisplatin 75 mg/m^2^ q21d 175 mg od for 2 days weeks 1–3 + gemcitabine 1000 mg/m^2^ weeks 1–3 (D1, D8 and D15) q28d	Established tolerable doses of oral Adavosertib in +carboplatin/cisplatin/gemcitabine that exceed threshold pharmacokinetic levels for efficacy and preliminary pharmacodynamic. The most common treatment-related AEs were Nausea (67%); Vomiting (35%); Diarrhea (41%); Fatigue (58%); Thrombocytopenia (44%); Neutropenia [32%]; Anemia [32%].

**Table 3 ijms-26-05701-t003:** List of WEE1 kinase inhibitors and their clinical/preclinical findings.

Drug	Sponsor	Mechanism and Key Properties	Clinical and Preclinical Highlights	References
Adavosertib	Astrazeneca, Cambridge, UK	Potent, selective ATP-competitive WEE1 inhibitor	Cytotoxic across tumor cell lines but limited clinical utility based on Phase I/II clinical trials	[89,150]
PD0166285	_	Pyridopyrimidine-based WEE1 inhibitor	Broad inhibition of other tyrosine kinases (CHK1, MYT1, c-Src, EGFR, and PDGFR), reducing clinical utility due to poor selectivity	[151,152,153]
PD0407824	_	WEE1 and CHK1 inhibitor	More selective than PD0166285 but less potent	[154]
Azenosertib	Zentalis Pharmaceuticals, New York, NY, USA	Oral WEE1 inhibitor with improved specificity and reduced toxicity compared to Adavosertib	Ongoing clinical trials in ovarian and uterine cancers	[113,155]
IMP7068	IMPACT Therapeutics, Nanjing, China	Highly selective WEE1 inhibitor (>435-fold selectivity over PLK1)	Well tolerated in Phase I trials for advanced solid tumors with no dose-limiting toxicities	[156,157]
SC-0191	Shijiazhuang Zhikang Hongren New Drug Development Co., Ltd., Shijiazhuang, China	WEE1 inhibitor	Demonstrated superior anti-tumor efficacy compared to Adavosertib in *TP53* mutant solid cancer preclinical studies	[158]
ATRN-1051	Aprea Therapeutics, Doylestown, PA, USA	Highly potent and selective WEE1 inhibitor	Effective at low doses in *CCNE1*-amplified ovarian cancer models, better pharmacokinetic properties than other WEE1 inhibitors	[159]

## 7. Conclusions and Future Directions

WEE1 kinase inhibitors hold potential as a novel therapeutic strategy for breast cancer, especially in tumors that rely on the G2/M checkpoint for DNA repair during replication. While significant advances have been made in preclinical research, the widespread clinical implementation of these inhibitors is yet to be achieved. This is mainly due to several challenges, including resistance to WEE1 inhibition, toxicities, and limited knowledge of predictive genetic biomarkers for response. Herein, we have covered the possible strategies to address these barriers in depth, including insights drawn from WEE1 inhibition studies in other cancers. We have compiled a detailed list of potential biomarkers that may predict treatment response. In addition to that, all combination therapeutic approaches that enhance cancer cell sensitivity to WEE1 inhibitors are listed in detail in this review, hoping it can aid in the design of more effective clinical trials in the future.

We would like to make a note that although Adavosertib can effectively decrease the growth of ER+ breast cancer cells that are resistant to antiestrogen and CDK4/6 inhibitors, the anti-proliferative effect of WEE1 inhibition in these cells is compromised when combined with antiestrogens or CDK4/6 inhibitors.

We would also like to emphasize that to fully harness the clinical potential of WEE1 inhibition, the development of next-generation WEE1 inhibitors with improved selectivity and reduced toxicity is crucial. Novel WEE1 inhibitors like ATRN-1051 (APR-1051), SC-0191, and Azenosertib have been developed by different pharmaceutical companies, and they may offer improved efficacy as well as safety over Adavosertib. Biomarker identification of therapeutic responses remains an unmet need. Combination therapy may be essential for successful clinical translation. Integrating WEE1 inhibition into precision medicine through biomarker-driven patient selection has the potential to transform breast cancer treatment and provide a potential treatment for patients with aggressive and therapy-resistant diseases.

## Figures and Tables

**Figure 2 ijms-26-05701-f002:**
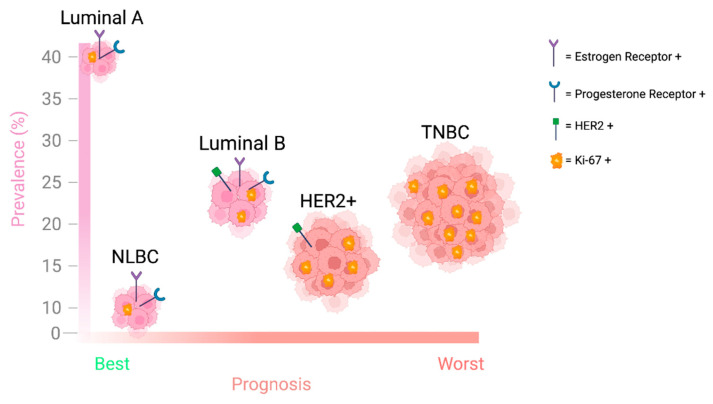
The figure describes the prognosis and the prevalence (% incidence) of the five phenotypic breast cancer subtypes. The size of the cancer cluster denotes the proportional contribution to cancer mortality for that subtype. Expression of key receptors in the progression of breast cancer and the proliferative potential (Ki67 positivity) of each subtype are also shown.

**Figure 3 ijms-26-05701-f003:**
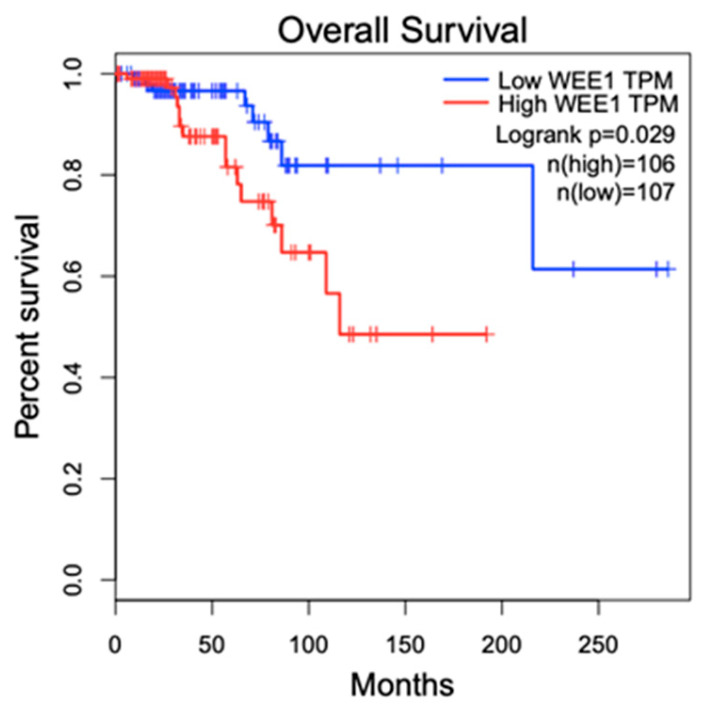
Kaplan–Meier plot from the TCGA BRCA database showing significant lower overall survival in WEE1-overexpressed breast cancer patients in comparison to patients with low WEE1 expression.

**Figure 4 ijms-26-05701-f004:**
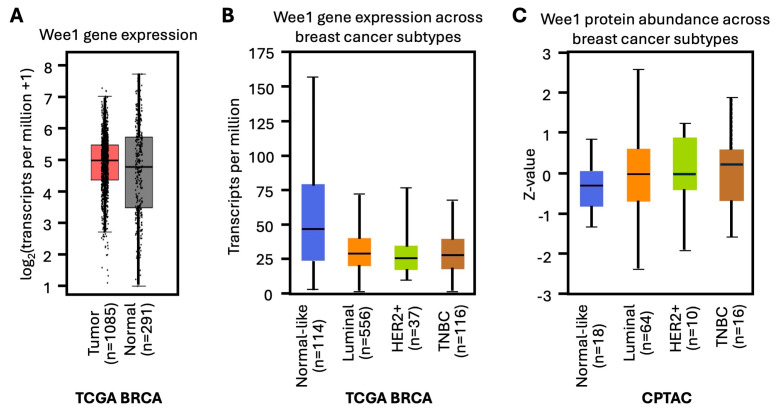
(**A**) Box plot showing comparison of WEE1 expression in normal breast tissue versus and breast cancer tissue samples (TCGA BRCA). (**B**,**C**) Box plots representing the gene and protein expression, respectively, of WEE1 in breast cancer patients based on the major subclasses. The analysis was conducted in TCGA BRCA cancer transcriptome data using the UALCAN database.

**Figure 5 ijms-26-05701-f005:**
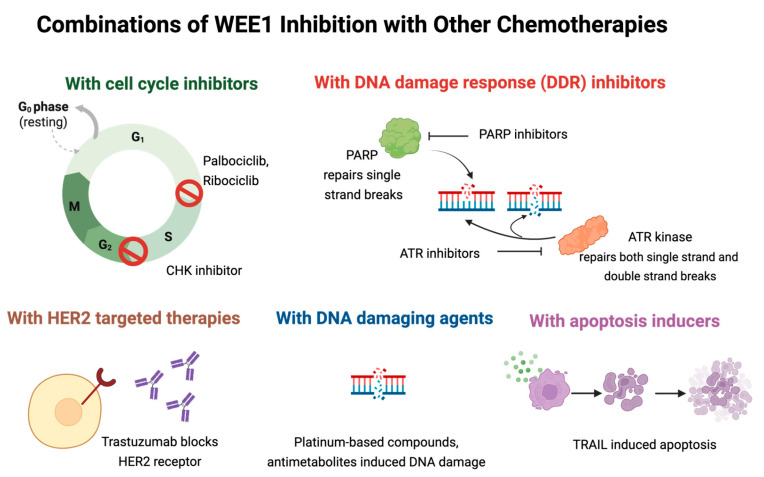
Schematic illustration of WEE1 inhibitors investigated in combination therapies during preclinical studies.

**Figure 6 ijms-26-05701-f006:**
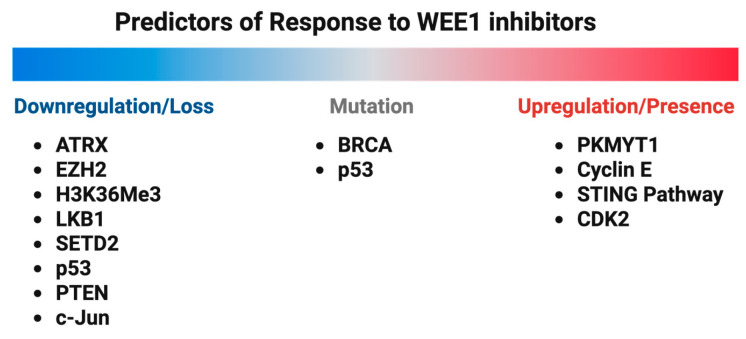
Schematic representation summarizing the predictors of response to WEE1 inhibitors.

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
