# Peer review of "Targeting WEE1 Kinase for Breast Cancer Therapeutics: An Update"

_ijms, 2025, doi:10.3390/ijms26125701_

Round 1

Reviewer 1 Report

Comments and Suggestions for Authors

This manuscript offers a comprehensive review of the most recent developments in WEE1 inhibitors for breast cancer therapy. The authors synthesize preclinical and clinical research on WEE1 inhibitors—particularly focusing on adavosertib—and explore combination strategies aimed at enhancing therapeutic efficacy. Their discussion of resistance mechanisms and predictive biomarkers, such as PKMYT1 upregulation and Cyclin E overexpression, underscores the clinical significance of WEE1 inhibition. Overall, this review is a valuable resource for appreciating the current landscape of WEE1-targeted treatments. I recommend publication following minor revisions. Notably, it would be advantageous to incorporate the most recent advances in WEE1 inhibitor research, including SGR-3515 and ACR-2316, both of which are in ongoing Phase I clinical trials.

Author Response

This manuscript offers a comprehensive review of the most recent developments in WEE1 inhibitors for breast cancer therapy. The authors synthesize preclinical and clinical research on WEE1 inhibitors—particularly focusing on adavosertib—and explore combination strategies aimed at enhancing therapeutic efficacy. Their discussion of resistance mechanisms and predictive biomarkers, such as PKMYT1 upregulation and Cyclin E overexpression, underscores the clinical significance of WEE1 inhibition. Overall, this review is a valuable resource for appreciating the current landscape of WEE1-targeted treatments. I recommend publication following minor revisions. Notably, it would be advantageous to incorporate the most recent advances in WEE1 inhibitor research, including SGR-3515 and ACR-2316, both of which are in ongoing Phase I clinical trials.

We appreciate the reviewer's comments.

Reviewer 2 Report

Comments and Suggestions for Authors

 The manuscript titled "Targeting WEE1 Kinase for Breast Cancer Therapeutics: An Update" presents a well-structured and informative narrative review summarizing the biological role of WEE1 kinase and its potential as a therapeutic target in various subtypes of breast cancer. The authors provide a comprehensive overview of WEE1’s involvement in DNA damage response, its dual oncogenic and tumor-suppressive functions, and current progress in preclinical and clinical development of WEE1 inhibitors. The inclusion of combination strategies and emerging resistance mechanisms is especially valuable. However, certain sections would benefit from deeper mechanistic discussion, improved clarity on clinical trial outcomes, and greater emphasis on biomarker-based patient stratification. Detailed comments are provided below to enhance the manuscript’s clarity, depth, and translational relevance.

Abstract

  1. The abstract effectively summarizes the review, but it lacks specificity. Consider adding key data points such as names of promising inhibitors (e.g., Adavosertib) and specific breast cancer subtypes (e.g., TNBC) to enhance clinical relevance.

Introduction

  1. This section provides a strong foundational background on WEE1’s role in cell cycle control. However, it would benefit from a graphical timeline or schematic summarizing the evolution of WEE1 as a therapeutic target, especially in cancer contexts.
  2. lines 65–70 discuss mitotic catastrophe but would be clearer if a reference to in vivo evidence of this mechanism were added.

 Lines 88–126 (Section 3: WEE1 in Breast Cancer)

  1. Clearly define which subtypes show upregulation vs downregulation of WEE1.
  2. Discuss whether WEE1 expression correlates with patient outcomes in clinical datasets (e.g., TCGA or METABRIC).
  3. Reference inconsistencies (e.g., Line 117–121) should include more data or a meta-analysis citation if available.

Lines 129–149 (Section 4: WEE1 Inhibitors)

  1. Clarify why Adavosertib, despite promising preclinical data, has struggled in breast cancer trials.
  2. Line 137: “over 2-fold” is vague. Add numerical tumor volume data or percent inhibition if available.

Lines 150–257 (Section 5: Combination Strategies)

  1. Provide exact percentage tumor regression in PDX models for clarity.
  2. Important contradiction—Adavosertib antiproliferative effects are compromised with CDK4/6 inhibitors. Emphasize this as a cautionary note in the conclusion.
  3. The upregulation of STING and MHC I is notable—consider adding these results to the abstract.

Lines 258–383 (Section 6: Resistance and Predictors)

  1. Cyclin E overexpression is clearly described. Add a visual summary linking each biomarker to its effect on WEE1 response.
  2. The paragraph on TP53 is conflicted. Suggest clearly stating: “TP53 mutation is not a reliable standalone biomarker for WEE1 inhibitor response.”

Lines 384–431 (Section 7: Clinical Trials)

  1. ORR of 26% “fell below the pre-decided cutoff” — specify what that cutoff was for transparency.
  2. List adverse events more succinctly in a table. Consider adding a risk-benefit matrix if space permits.

Lines 434–450 (Emerging WEE1 Inhibitors)

1.Comparing IC50 values and pharmacodynamic properties across Adavosertib, ZN-c3, SC-0191, and ATRN-1051.

  1. Highlighting why these agents may outperform first-generation inhibitors in breast cancer specifically.

Lines 452–472 (Conclusion and Future Directions)

  1. Suggest reiterating that combination therapy and biomarker selection are essential for successful clinical translation.
  2. Line 471: Consider softening “renewed hope” to a more neutral scientific tone, such as “potential for improved outcomes.”

Author Response

Reviewer 2

The manuscript titled "Targeting WEE1 Kinase for Breast Cancer Therapeutics: An Update" presents a well-structured and informative narrative review summarizing the biological role of WEE1 kinase and its potential as a therapeutic target in various subtypes of breast cancer. The authors provide a comprehensive overview of WEE1’s involvement in DNA damage response, its dual oncogenic and tumor-suppressive functions, and current progress in preclinical and clinical development of WEE1 inhibitors. The inclusion of combination strategies and emerging resistance mechanisms is especially valuable. However, certain sections would benefit from deeper mechanistic discussion, improved clarity on clinical trial outcomes, and greater emphasis on biomarker-based patient stratification. Detailed comments are provided below to enhance the manuscript’s clarity, depth, and translational relevance.

Abstract

  1. The abstract effectively summarizes the review, but it lacks specificity. Consider adding key data points such as names of promising inhibitors (e.g., Adavosertib) and specific breast cancer subtypes (e.g., TNBC) to enhance clinical relevance.

            The names of promising inhibitors and specific breast cancer subtypes have been added to enhance clinical relevance.

Introduction

  1. This section provides a strong foundational background on WEE1’s role in cell cycle control. However, it would benefit from a graphical timeline or schematic summarizing the evolution of WEE1 as a therapeutic target, especially in cancer contexts. 

A graphical timeline has been inserted.

  1. lines 65–70 discuss mitotic catastrophe but would be clearer if a reference to in vivo evidence of this mechanism were added.

We added the new reference into the manuscript (Mir., S Cancer Cell )

 Lines 88–126 (Section 3: WEE1 in Breast Cancer)

  1. Clearly define which subtypes show upregulation vs downregulation of WEE1.

We have included a breast cancer subset analysis (Fig. 4) (WEE1 is expressed at high levels in several cancer types, including luminal and HER-2-positive breast cancer).

Discuss whether WEE1 expression correlates with patient outcomes in clinical datasets (e.g., TCGA or METABRIC).

We conducted new analysis for correlation between overall survival and WEE1 expression level based on the public database and included the results in the revised manuscript.

  1.  

Reference inconsistencies (e.g., Line 117–121) should include more data or a meta-analysis citation if available.

We corrected the references, added new reference to the correlation of high level WEE1 expression with poor prognosis in TNBCs and the overall survival analysis by ourselves based on TCGA breast cancer dataset which showed high level WEE1 expression corelates with poor survival. 

  1.  

Lines 129–149 (Section 4: WEE1 Inhibitors)

  1. Clarify why Adavosertib, despite promising preclinical data, has struggled in breast cancer trials.
  1. We have clarified that toxicity and lack of genetic predictors have been a barrier to clinical translation of WEE1 inhibitors.
  1. Line 137: “over 2-fold” is vague. Add numerical tumor volume data or percent inhibition if available.

We deleted “over 2-fold” and added detail information as: “The IC50 of Flavokawain A on the growth of HER2 overexpressing SKBR3 and MCF-7/HER2 cells are 10 and 13.6 μM, respectively, versus 38.4 and 45 μM for MCF7 and MDA-MB-468 cells, respectively. Flavokawain A at a concentration of 4 μM inhibits the colony formation of MCF/HER2 and MCF7 by 80% and 54%, respectively.

Lines 150–257 (Section 5: Combination Strategies)

  1. Provide exact percentage tumor regression in PDX models for clarity.

The exact percentage wasn’t mentioned in the publication. We calculated from the figures in the publication and added “Compared to Dinaciclib monotherapy, the sequential combination treatment with Adavosertib reduced the tumor volume by 60% in PDX models”.

  1. Important contradiction—Adavosertib antiproliferative effects are compromised with CDK4/6 inhibitors. Emphasize this as a cautionary note in the conclusion.

We added “We would like to make a note that although Adavosertib can effectively decrease the growth of ER+ breast cancer cells that are resistant to antiestrogen and CDK4/6 inhibitors, the anti-proliferative effect of WEE1 inhibition in these cells is compromised when combined with antiestrogens or CDK4/6 inhibitors.” in the conclusion.

  1. The upregulation of STING and MHC I is notable—consider adding these results to the abstract.

Upregulation of STING and MHC I when added in preclinical combination therapies was added to the abstract.

Lines 258–383 (Section 6: Resistance and Predictors)

  1. Cyclin E overexpression is clearly described. Add a visual summary linking each biomarker to its effect on WEE1 response.
  1. Figure 5 has been added to the manuscript.
  1. The paragraph on TP53 is conflicted. Suggest clearly stating: “TP53 mutation is not a reliable standalone biomarker for WEE1 inhibitor response.”

The text was changed in keeping with the reviewer’s suggestion.

Lines 384–431 (Section 7: Clinical Trials)

  1. ORR of 26% “fell below the pre-decided cutoff” — specify what that cutoff was for transparency.

The cutoff of 30% is now mentioned in the Table.

  1. List adverse events more succinctly in a table. Consider adding a risk-benefit matrix if space permits.

The adverse effects are now mentioned in the table, although there is insufficient information to create a risk-benefit matrix.

Lines 434–450 (Emerging WEE1 Inhibitors)

1.Comparing IC50 values and pharmacodynamic properties across Adavosertib, ZN-c3, SC-0191, and ATRN-1051.

We added the “Additional WEE1 kinase inhibitors and their IC50 for WEE1 kinase inhibition include Azenosertib (ZN-c3, IC50 of 3.9 nM), SC0191 (IC50 of 22.3), ATRN-1051 (APR-1051, IC50 of 2.25 nM)  IMP7068 (WEE1-IN-10, Potrasertib, IC50 of 0.524 μM), PD0407824 (Chk1 and WEE1 inhibitor with IC50 of 47 and 97 nM), PD0166285 (IC50 of 24 nM), WEE1- IN -5 (IC50 of 0.8 nM), De bio123 (Zedoresertib, IC50 of 0.8 nM) and WEE1-IN-8 (IC50 of 0.98 nM) have been developed.” And the clinical trials for other WEE1 kinase inhibitors include Azenosertib (NCT04814108, NCT04972422, NCT04158336), SC0191 (NCT06363552), ATRN-1051 (NCT06260514), IMP7068 (NCT04768868), Debio123 (NCT03968653, NCT04855656) are in clinical trials.

  1. Highlighting why these agents may outperform first-generation inhibitors in breast cancer specifically.

We have added the limited known information to speculate why these agents may prove effective.

Lines 452–472 (Conclusion and Future Directions)

  1. Suggest reiterating that combination therapy and biomarker selection are essential for successful clinical translation.

We have reiterated that combination therapy and biomarker selection are essential for successful clinical translation in the Conclusion and Future Directions

  1. Line 471: Consider softening “renewed hope” to a more neutral scientific tone, such as “potential for improved outcomes.”

We made the change based on the Reviewer’s comments.

Reviewer 3 Report

Comments and Suggestions for Authors

Zhang et al present a comprehensive review of the current and future prospects for the use of Wee1 kinase inhibitors in the treatment of Breast Cancer.  The review begins with an overview of the molecular functions of Wee1 kinase in cell cycle control, its role in implementing cell cycle  checkpoint arrest under conditions of DNA damage, and its capacity to maintain normal DNA replication rates through regulation of CDK2.  This is followed by a review of the different phenotypic categories of Breast Cancers, their genetic heterogeneity, conventional treatments, and their respective prognostic outlooks.  Building on this the authors then consider the preclinical evidence for Wee1 inhibitor (in practice almost exclusively Adavertosib also known as AZD1775) anticancer activity either as a single agent or in combination with a variety of classic chemotherapeutics or molecularly targeted agents.  Consideration is then given to other genetic abnormalities common in Breast Cancers (and a few other cancer types) as potential agents of resistance or as predictors of response.  The review finishes with an overview of the outcomes of various clinical trials performed thus far using Adavertosib, which in the main have been somewhat disappointing, together with an evaluation of the forthcoming next generation of Wee1 inhibitors which are hoped to exhibit greater selectivity and potency.

In general the review is well-structured and compiles a large body of information likely to be of interest and use to cancer pharmacologists and clinicians.  There are a number of small points which if corrected would improve the ms.

1) Lines 39-40 – Wee1 is described as a “serine-threonine kinase” which of course is true, yet the only functional substrate sites discussed in the article are the inhibitory tyrosine residues in CDK1 and CDK2.  This might confuse some readers so a little further explanation regarding the very unusual situation presented by Wee1 seems warranted.

2) Fig. 2 – legend states that “prognosis (months of survival)” are illustrated but the figure merely shows “Best to Worst” on the prognosis axis.

3) Lines 135-141 – reference to quercetin and other bioflavonoids as possible inhibitors of Wee1 seems largely superfluous and could be removed.

4) Fig. 3, middle upper panel indicating “DDR inhibitors”.  It is difficult to understand what this panel seeks to convey.  Presumably a DNA double strand break, but any molecular biologist looking at this would see the left hand sequence is an EcoR1 site (GAATTC), whilst the sequence on the right hand presents a cytosine mispaired with a cytosine, which is clearly incorrect.  Not at all clear how this image is intended to convey the consequences of ATR, PARP inhibition.

5) Line 274 – “Cyclin E inhibits CDK2” is counter to conventional wisdom that CDK2 gains catalytic activity when associated with cyclin E.  Mention is also made of “a higher prevalence of cyclin E mutations” however it is a little unclear whether these affect cyclin E expression, by gene amplification for example, or do they in fact affect the protein coding sequence of cyclin E.  If the latter, it would seem appropriate to say something about the functional consequences of said mutations.

6) Lines 312-313 – “Wee1 inhibition……reduces RRM2 reduction” very unclear what is meant here by this double negative – needs to be reworded more clearly.

7) Terminoloy – desirable to use consistent terms throughout.  Presumably AZD1775 was renamed Adavertosib for clinical, as opposed to preclinical use, so maybe best to use that convention throughout.  Another example, NCT03012477 refers to evaluation of “pCDC2” as a biomarker – presumably this is a cut and paste error, but the correct terminology is pCDK1, which is otherwise used throughout.

Author Response

Reviewer 3

Zhang et al present a comprehensive review of the current and future prospects for the use of Wee1 kinase inhibitors in the treatment of Breast Cancer.  The review begins with an overview of the molecular functions of Wee1 kinase in cell cycle control, its role in implementing cell cycle  checkpoint arrest under conditions of DNA damage, and its capacity to maintain normal DNA replication rates through regulation of CDK2.  This is followed by a review of the different phenotypic categories of Breast Cancers, their genetic heterogeneity, conventional treatments, and their respective prognostic outlooks.  Building on this the authors then consider the preclinical evidence for Wee1 inhibitor (in practice almost exclusively Adavertosib also known as AZD1775) anticancer activity either as a single agent or in combination with a variety of classic chemotherapeutics or molecularly targeted agents.  Consideration is then given to other genetic abnormalities common in Breast Cancers (and a few other cancer types) as potential agents of resistance or as predictors of response.  The review finishes with an overview of the outcomes of various clinical trials performed thus far using Adavertosib, which in the main have been somewhat disappointing, together with an evaluation of the forthcoming next generation of Wee1 inhibitors which are hoped to exhibit greater selectivity and potency.

In general the review is well-structured and compiles a large body of information likely to be of interest and use to cancer pharmacologists and clinicians.  There are a number of small points which if corrected would improve the ms.

1) Lines 39-40 – Wee1 is described as a “serine-threonine kinase” which of course is true, yet the only functional substrate sites discussed in the article are the inhibitory tyrosine residues in CDK1 and CDK2.  This might confuse some readers so a little further explanation regarding the very unusual situation presented by Wee1 seems warranted. 

We have replaced the text referring to the tyrosine kinase activity for the inhibitory tyrosine residues in CDK1 and CDK2.

2) Fig. 2 – legend states that “prognosis (months of survival)” are illustrated but the figure merely shows “Best to Worst” on the prognosis axis.

We deleted months of survival from the figure legend as requested.

3) Lines 135-141 – reference to quercetin and other bioflavonoids as possible inhibitors of Wee1 seems largely superfluous and could be removed.

We have taken the liberty of keeping this paragraph out of respect to the authors of these prior studies that provided rationale that the patients could benefit from WEE1 inhibition.

4) Fig. 3, middle upper panel indicating “DDR inhibitors”.  It is difficult to understand what this panel seeks to convey.  Presumably a DNA double strand break, but any molecular biologist looking at this would see the left hand sequence is an EcoR1 site (GAATTC), whilst the sequence on the right hand presents a cytosine mispaired with a cytosine, which is clearly incorrect.  Not at all clear how this image is intended to convey the consequences of ATR, PARP inhibition.

A new image was made with a better display of mechanism is now included as Figure 4.

5) Line 274 – “Cyclin E inhibits CDK2” is counter to conventional wisdom that CDK2 gains catalytic activity when associated with cyclin E.  Mention is also made of “a higher prevalence of cyclin E mutations” however it is a little unclear whether these affect cyclin E expression, by gene amplification for example, or do they in fact affect the protein coding sequence of cyclin E.  If the latter, it would seem appropriate to say something about the functional consequences of said mutations.

We corrected to “Cyclin E activates CDK2”.

6) Lines 312-313 – “Wee1 inhibition……reduces RRM2 reduction” very unclear what is meant here by this double negative – needs to be reworded more clearly.

The sentence was reworded as “WEE1 inhibition in H3K36me3-deficient cells rescues RRM2 expression, leading to dNTP depletion, S-phase arrest, and apoptosis.”

7) Terminoloy – desirable to use consistent terms throughout.  Presumably AZD1775 was renamed Adavertosib for clinical, as opposed to preclinical use, so maybe best to use that convention throughout.  Another example, NCT03012477 refers to evaluation of “pCDC2” as a biomarker – presumably this is a cut and paste error, but the correct terminology is pCDK1, which is otherwise used throughout.

We made the changes throughout based on the Reviewers suggestion.

Round 2

Reviewer 2 Report

Comments and Suggestions for Authors

The authors have adequately addressed my previous comments. I have no further concerns, and the revised manuscript is acceptable for publication in its current form.